# Confirmatory Factor Analysis of Comorbidity between Depression and Aggression in a Child-Adolescent Community Sample: Nosological, Prognosis and Etiological Implications

**DOI:** 10.3390/ijerph19084424

**Published:** 2022-04-07

**Authors:** Rodolfo Gordillo, María Victoria Del Barrio Gándara, Miguel A. Carrasco

**Affiliations:** 1Department of Psychology and Criminology, University at Distance of Madrid (UDIMA), C/Camino de la Fonda 20, 28400 Collado Villalba, Spain; 2Department of Personality, Psychological Evaluation and Treatment, Universidad Nacional de Educación a Distancia (UNED), C/ de Juan del Rosal, 10, 28040 Madrid, Spain; vbarrio@psi.uned.es (M.V.D.B.G.); macarrasco@psi.uned.es (M.A.C.)

**Keywords:** comorbidity, depression, aggression, childhood, adolescence, longitudinal, SEM

## Abstract

Comorbidity between depression and aggression in the child-adolescent population remains a controversial phenomenon. To our knowledge, no longitudinal study using structural equation modeling (SEM) has confirmed whether the relationship between depression and aggression is due to the fact that they share internalizing and externalizing supraordinal factors at the level of the syndrome or is due to the fact that they share common characteristics in relation to an underlying factor at the level of symptoms. We examined longitudinal comorbid relationships in a community sample (N = 251) at three waves ages from 10 to 13 years. The SEM showed that longitudinally, the comorbidity between depression and aggression is due to the fact that they share characteristics of the same underlying factor at the symptom level. These results have implications for the classification, diagnosis, and treatment of comorbidity between depression and aggression in a child-adolescent population.

## 1. Introduction

Comorbidity between two psychological disorders is characterized by a synergistic effect that clinically and significantly alters the intensity of the core symptoms of both disorders, beyond what the expression of each of them would have independently [1,2,3]. Comorbidity can occur between internalizing or externalizing disorders called homotypic, as well as between internalizing and externalizing disorders called heterotypic [4,5]. This last type of comorbidity is a complex phenomenon to prove because the synergistic effect occurs between two a priori divergent disorders [6] defined by contradictory symptoms [7]. In this regard, the comorbidity between depression and aggression presents a great epistemological challenge for the scientific community [8,9]. According to the diagnostic standards of the categorical methodology, comorbidity should not exist [10].

Clinical evidence, as well as quantitative studies, show that psychopathological comorbidity in children and adolescents not only exists [11] but is the rule more than the exception [12,13]. Data from Angold and Costello’s meta-analysis in late 1990 [4] indicated that both homotypic and heterotypic comorbidity occurred with an odds ratio of 3 to 10.7. Likewise, 20 years later, Essau and de la Torre-Luque [9] reported that the prevalence of comorbidity in the child-adolescent population was as high as 60%. In this line, rates of heterotypic comorbidity between depression and aggression, two of the disorders that require more clinical attention in the child and adolescent population [14,15], range from 8–11% in the general population [16,17], increasing to 24% in the clinical population [18]. These high rates suggest important problems with current nosology [19], which poses different problems for obtaining useful clinical care [20]. Thus, child and adolescent studies have found that the effect of comorbidity increases: (a) the symptomatic duration of comorbid disorders [1]; (b) the failure of standard interventions [2,21,22,23,24] and (c) the spurious results in clinical improvements [25]. 

The distinction between comorbidity successive or episodic and concurrent or chronic is key for improving its diagnosis [4,26]. The increase in knowledge of comorbidity is in the study of the concurrent type [20]. According to Angold, two comorbid disorders over time also share phenomenology or cause [27]. The problem of concurrent comorbidity is that it can only be evaluated through longitudinal studies [7,28] and comorbidity between depression and aggression, although correlational studies have shown its prospective association [18,29], there is a lack of confirmatory studies that have longitudinally verified its phenomenological stability [30]. 

To date, only Weiss and Catron with a cross-sectional design have studied the phenomenology of comorbidity between depression and aggression in children and adolescents [31]. These authors found that the cause of the comorbidity between depression and aggression is plausible both at the broadband and at the narrowband level. Thus, for Weiss and Catron the treatment of depressed and aggressive comorbid children and adolescents may focus on both negative emotionality (a generalized disposition to experience broad-spectrum unpleasant affective states) [32] and irritability [33], a symptom they share specifically at narrowband level [34]. These findings are ambiguous and do not allow us to improve our knowledge of the etiology and treatment of comorbidity between depression and aggression in children and adolescents.

In short, longitudinal studies are needed to confirm whether comorbidity between depression and aggression is successive or concurrent in the child-adolescent population, as well as whether its phenomenology remains stable at the broadband level, or more specifically at the narrowband level. Therefore, with the aim of improving scientific knowledge of epistemological, prognostic and nosological aspects [35], this longitudinal study carried out in a general sample of children and adolescents aged from 9–15 years, hypothesized that the comorbidity between depression and aggression is stable across time or concurrent as a cause of what they share both at the broadband level and at the narrowband level.

## 2. Materials and Methods

### 2.1. Participants

The present study analyzed longitudinal data from 525 children and adolescents in the region of Madrid. The mean age of the participants was 10.86 at the first wave (Time 1; T1), 11.86 at the second wave (Time 2; T2), and 12.86 at the third wave (Time 3; T3), ranging from 9 to 15 years. The proportion of respondents with respect to the total number of participants who started the study was 47.5% (T1), 45.3% (T2) and 53.4% (T3). Finally, for this study, participants were 251 (167 girls and 84 boys) who completed those measures on depression and physical and verbal aggression at T1, T2, and T3.

To obtain a representative sample we used the formula: n = Z^2^ α.p.q/d^2^ [36]. According to the official census and considering the estimated proportion by sex (47.7%), the study required 70 participants. If we estimate the sample size according to the prevalence of comorbidity between depression and aggression in the child-adolescent population [3], the study required 140 participants. Therefore, a sample of 251 participants adequately represents the child-adolescent population of Madrid, with a confidence level of 95% and a sampling error of 5%. In summary, the sample size of this study allows us to generalize the results obtained about the plausibility of the proposed model.

### 2.2. Procedure

The selection of the sample was carried out by random probabilistic sampling. Age and sex were taken as population parameters. This study focused on the set of public and private schools provided by the Education Delegation of the Community of Madrid. After contacting the director of the center and commenting on the proposal for data collection, we sent them a summary of the goals of the study and formally requested their participation in the research, as well as their authorization. If the answer was affirmative, the school was selected for data collection. This same procedure was carried out in 15 schools. Finally, there were 10 who agreed to participate in the study.

Planning was conducted jointly by a team of psychologists trained for the task and the teachers in charge of each classroom. The tests were conducted by the psychologists in the class groups with a maximum of 25 students per classroom. Before the beginning of the evaluation, the aims of the research and the evaluation instruments were briefly explained to the subjects. The Ethics Commission of the National University of Distance Education (UNED) authorized the study and its compliance with the ethical and data protection standards required by European legislation.

### 2.3. Instruments

Depression. Participants completed the Childhood Depression Inventory, Short Version (CDI-S; [37]). Spanish adaptation by del Barrio, Roa [38]. The short form is a self-report measure with 10 items written at a 7-year-old reading level. The items represent three levels of depressive intensity on the emotional state experienced during the past two weeks in negative self-esteem (e.g., “Nobody loves me”), anhedonia (e.g., “I am always sad”), and hopelessness (e.g., “I’ll never do anything right”). Internal consistency by Cronbach’s alpha obtained in the US population was 0.79 [37]. In the Spanish adaptation, Cronbach’s alpha was 0.71 [38]. In this study, Cronbach’s alpha was 0.73, 0.73, and 0.74 at T1, T2, and T3, respectively. The construct validity of the CDI-S with the Self-Esteem Scale (GSES) was *r_xy_* = 0.64 (*p* < 0.001) and with the State-Trait Anxiety for Children (STAIC) were *r_xy_* = 0.65 and *r_xy_* = 0.62 (*p* < 0.0001) for state and trait anxiety, respectively [38].

Physical and verbal aggression was measured using the Physical and Verbal Aggression Questionnaire (FVA; [39]). Spanish adaptation by del Barrio, Moreno [40]. The FVA consists of 20 items that measure different aggressive behaviors, both physical (e.g., “Kicking and punching”) and verbal (e.g., “Cursing”, “Threatening others”). The response format is 3 alternatives (*often*, *sometimes* or *never*). Its application is individual or collective for children and adolescents aged from 7 to 17 years. The Cronbach’s alpha of the original instrument is 0.86 [39]. In the Spanish adaptation, Cronbach’s alpha was 0.84 [40]. Cronbach’s alpha for this investigation was 0.81, 0.83, and 0.8 (physical aggression) and 0.75, 0.78, and 0.74 (verbal aggression) at T1, T2, and T3, respectively. The construct validity of the FVA with the Children Depression Inventory is *r_xy_* = 0.4 (*p* < 0.001) and *r_xy_* = 0.09 (*p* < 0.05) with the Emotional Instability Inventory [40].

### 2.4. Statistical Analysis

For the development of this study, we estimated three longitudinal structural equation models (SEM-L1, L2 and L3). First, with SEM-L1 we tested that phenomenologically, depression and aggression co-occur significantly across time. Second, with SEM-L2 we examined whether it is plausible that broadband comorbidity is chronic or stable. Finally, with SEM-L3 we tested whether it was plausible that the comorbidity between depression and aggression at the narrowband level was stable across time or chronic.

We ran the parameters of the SEM models using the maximum likelihood method (MMP). This method allows estimating the adequacy of the model from the Chi-square statistic [41]. Its value should indicate that it is not significant, but it is too sensitive to the sample size, resulting in a tendency to reject the null hypothesis [42,43]. Therefore, in addition to the Chi-square, we used three standard fit indexes to test our SEM models: (a) the Root Mean-square Error of Approximation (RMSE; [44]), which requires a value less than 0.05 for the acceptance of the proposed model, although values that do not exceed 0.08 represent reasonable errors of approximation of the model to the population [45]; (b) the Comparative Fit Index (*CFI*; [46]), that has to reach values close to 0.95 and (c) Hoelter‘s critical *N* [47], which test the adequacy of the sample size and must be greater than 200. 

In addition, due to the controversy over the choice of standard fit indexes, we incorporated the Standardized Root Mean Square Residual (*SRMR*) into the analysis, as recommended by Mai et al. [48]. These authors have designed a decision tree on SEM fit indexes according to three parameters: (1) what is the purpose of the study; (2) what is the focus of the estimate and (3) what is the sample size. In this sense, our study proposes to examine a new longitudinal model with structural equation analysis and with a sample size greater than 200. According to the decision tree, the most appropriate fit index for our research is the *SRMR*, which requires values lower than 0.08. 

The analysis of structural equation models (SEM) was conducted using AMOS program version 18.

## 3. Results

SEM-L1 had an adequate fit of the data (*χ*^2^(10) = 23.903, *p* < 0.05; *CFI* = 0.99, *RMSEA* = 0.07, *Hoelter’s Critical* = 251, *SRMR* = 0.0618). As predicted, these results indicate that a model represented by longitudinal comorbidity between depression and aggression is plausible (Figure 1). The correlation between these two constructs was 0.29, showing a small but sustained co-occurrence across time. Therefore, comorbidity between depression and aggression is a plausible phenomenon that remains stable in children and adolescents aged 10.86 to 12.86 years.

SEM-L2 did not fit the data (*RMSEA* = 0.41). This result was different from what was expected, so we tried different changes of parameters to fit the model. We constrained variances of the measurement errors to have the same value. Additionally, we equalized the regression weights in different trajectories from the latent factors to the observable variables. All new models were null or overidentified. Therefore, we rejected the search for a fit that did not correspond to the original theory and structure developed by Weiss and Catron. In short, these findings indicate that the longitudinal comorbidity between depression and aggression in a child-adolescent sample aged 10.86 to 12.86 years, does not occur at the broadband or syndrome level, through the relationship between the dimensions of second internalizing and externalizing order (Figure 2).

Finally, SEM-L3 had an adequate fit of the data (*χ*^2^ (10) = 23.903, *p* = 0.008; *CFI* = 0.99; *RMSEA* = 0.07; *Hoelter’s Critical* = 251, *SRMR* = 0.0618). Interestingly, the findings were identical to meet for the SEM-L1 (Figure 3). This assumes that no adjustment had to be made to identify the data and thus artificially bias the model identification. Therefore, these findings indicate that the longitudinal comorbidity between depression and aggression in a child-adolescent sample aged from 10.86 to 12.86 years, occurs at the narrowband level (Figure 3). That is, depression and aggression are correlated by specific symptoms that they share in the age range measured. Such unique symptoms explain a shared comorbidity variance of 28% and 31% between depression and aggression, respectively.

## 4. Discussion

The study of comorbidity in the child and adolescent population, with prevalence as high as 60%, is essential to achieve advances in nosology, prognostic and etiology of this phenomenon [2,11]. In this regard, the study of comorbidity between depression and aggression lacks SEM research. Hence, it is essential to develop longitudinal research designs and strategies that prove both its existence and the degree of expression of these comorbid disorders [19]. Consequently, we developed three plausible models (SEM-L1, L2 and L3) to test the longitudinal relationship between depression and aggression in a child-adolescent community sample measured in three waves (10.86, 11.86 and 12.86 years). We hypothesize that the cause of comorbidity between depression and aggression is stable across time or concurrent, due to what they share at both the broadband and narrowband levels. In SEM-L1, we tested whether the comorbidity between depression and aggression is stable across time, specifically, from 10 to 13 years of age. Next, we analyzed with SEM-L2 whether the cross-sectional model proposed by Weiss and Catron would fit the data longitudinally [31]. Finally, with SEM-L3 we longitudinally investigated whether the comorbidity between both disorders was due to a shared variance at the level of symptoms or narrowband.

The findings show that the SEM-L1 and SEM-L3 models had a good fit for the data. Therefore, one of the hypotheses confirmed by this study was that comorbidity between depression and aggression in a child-adolescent sample was concurrent or chronic. Following Eaton [19], we agree with the importance of its detection, as well as the degree and/or intensity of its manifestation. Thus, our study found that comorbidity between depression and aggression with a correlation of 0.29 was chronic from 10.86 to 12.86 years.

The analysis of the SEM-L2 that emulated that of Weiss and Catron [31], shows that it did not fit our data. Therefore, we were unable to longitudinally confirm the existence of two broadband syndromes as a cause of the comorbid relationship between depression and aggression in a child-adolescent sample. This may be due to the fact that our model, unlike the one presented by these authors, did not include in the model either anxiety as part of the internalizing broadband syndrome, or hyperactivity as part of the externalizing broadband dimension syndrome. According to Weiss and Catron, the comorbidity between depression and aggression is due to the shared effect with the other components of the internalizing (anxiety) and externalizing (hyperactivity) dimensions. Therefore, our results add to the controversy shown by McDonough-Caplan, and Klein [5]. These authors state that in recent years it has become clear that attributing any comorbid effect in childhood and adolescence to what they share in the syndromes, internalizing and externalizing is not always justified. Evidence indicates that transdiagnostic psychopathological vulnerabilities extend both within and across broadband spectra [49]. Future research will have to continue addressing this issue to find out how much both comorbid disorders share both homotypic and heterotypic.

Finally, SEM-L3 findings showed that heterotypic comorbidity between depression and aggression in children and adolescents occurs through the vulnerability they share specifically at the symptom level. This result confirms our hypothesis about the existence of a shared variance due to common symptomatology of depression and aggression at the narrowband level. This longitudinal finding of comorbidity between depression and aggression has important significance for treatment and prevention. As stated by Dutton and Karakanta [7], it is a mistake to think that their apparent contradiction makes them divergent, since there is evidence that the presence of depression increases the risk of general aggression, aggression towards intimate partners and self-injury. For these authors, there is a common characteristic at the symptom level, irritability. Likewise, Zisner and Beauchaine [50] also found a shared heritability that supports the common vulnerability in heterotypic comorbidity between depression and externalizing syndrome. Specifically, these authors found genetic support for a trait characterized by a response pattern toward impulsivity, anhedonia, and irritability as a result of shared variance between depression and externalizing syndrome. Therefore, the findings of this study contribute to the growing support for the transdiagnostic approach. Consequently, the high relapse rate in conventional treatment for depression and aggression [51] needs to be reassessed in relation to their shared comorbid variance.

This study has some limitations to consider. First, our data were based on child and adolescent self-reports and did not incorporate data from other informants or direct observations of them. Future research should incorporate data from multiple sources. Second, it would be convenient for further research to extend the study periods from younger ages to adulthood, as well as to explore these relationships with different dimensions of the depression construct (e.g., anhedonia, dysphoria, low self-esteem) and different expressions of aggression beyond physical and verbal aggression. Third, for theoretical reasons, the findings of the present study should be limited to the general symptoms of depression and aggression (physical and verbal) from a dimensional approach. Finally, gender differences have not been analyzed in this study, hence, this is an important limitation since it is well known, that aggression tends to be higher in boys and depression in girls. Future studies should analyze the longitudinal invariance by gender.

## 5. Conclusions

The present investigation through SEM analysis showed the importance of conducting longitudinal studies on the comorbidity between depression and aggression in a child-juvenile community sample. Their relationship at the symptom level indicates a longitudinally significant co-occurrence. This finding suggests that their study only at the syndrome level (internalized and externalized) could mask the specific variance shared by comorbid disorders. Therefore, it will be difficult to clarify which symptoms are specific, which are common, and most importantly when can they be detected and how they develop across time [51]. Undoubtedly, these responses have direct implications for improving the etiology, prognosis and psychopathological nosology in childhood and adolescence. In this respect, this study found a chronic comorbid structure at symptom level between depression and aggression, from 10 to 13 years.

## Figures and Tables

**Figure 1 ijerph-19-04424-f001:**
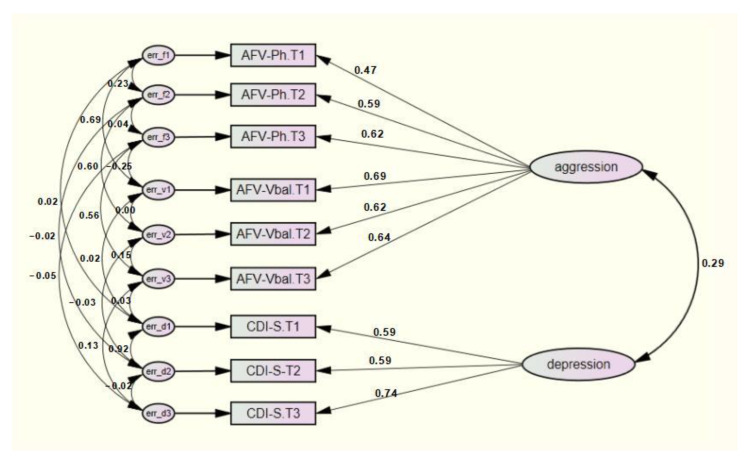
SEM-L1 model represented by longitudinal comorbidity between depression and aggression with standardized coefficients. Measurements moments: T1 = 10.86 years old, T2 = 11.86 years old, T3 = 12.86 years old. Latent factors: aggression and depression. Observed variables: AFV-Ph = Physical and Verbal Aggression Questionnaire, items physical aggression; AFV-Vbal = Physical and Verbal Aggression Questionnaire, items verbal aggression; CDI-S = Children Depression Inventory-Short version.

**Figure 2 ijerph-19-04424-f002:**
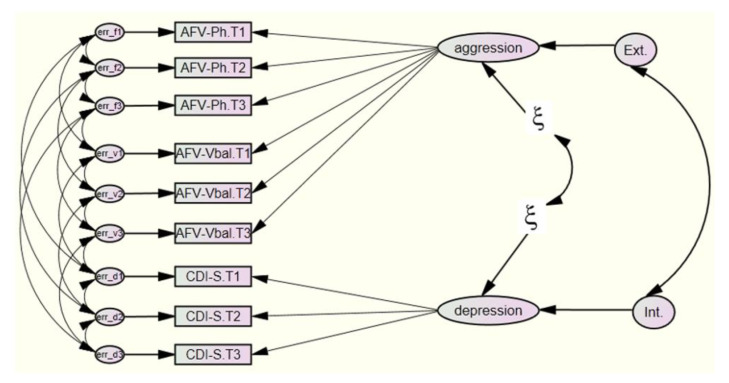
SEM-L2 model represented by second-order confirmatory factors at the broadband level. Structural model L2. Measurements moments: T1 = 10.86 years old, T2 = 11.86 years old, T3 = 12.86 years old. Latent factors: aggression, depression (first order factors), Ext. = Externalizing, Int. = Internalizing (second order factors). Observed variables: AFV-Ph = Physical and Verbal Aggression Questionnaire, items physical aggression; AFV-Vbal = Physical and Verbal Aggression Questionnaire, items verbal aggression; CDI-S = Children Depression Inventory-Short version).

**Figure 3 ijerph-19-04424-f003:**
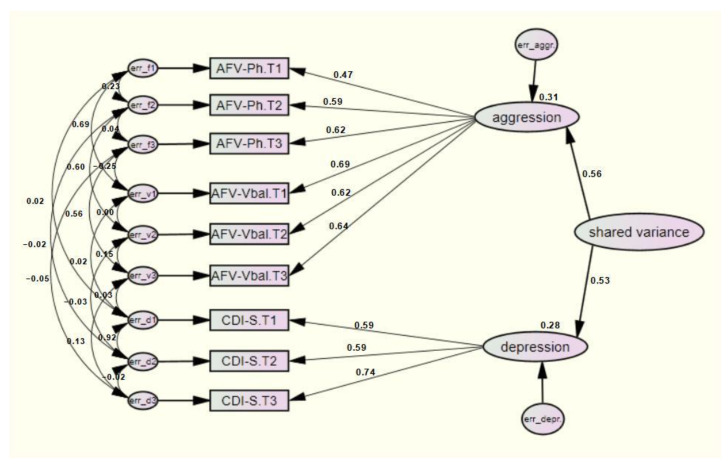
SEM-L3 model represented by the narrow-band level. Structural model L3 with standardized coefficients. Measurements moments: T1 = 10.86 years old, T2 = 11.86 years old, T3 = 12.86 years old. Latent factors, aggression, depression (first order factors), and shared variance (second order factor). Observed variables: AFV-Ph = Physical and Verbal Aggression Questionnaire, items physical aggression; AFV-Vbal = Physical and Verbal Aggression Questionnaire, items verbal aggression; CDI-S = Children Depression Inventory-Short version).

## Data Availability

The dataset generated during and/or analyzed during the current study is available from the corresponding author upon reasonable request.

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
