# Peer review of "Confirmatory Factor Analysis of Comorbidity between Depression and Aggression in a Child-Adolescent Community Sample: Nosological, Prognosis and Etiological Implications"

_ijerph, 2022, doi:10.3390/ijerph19084424_

Round 1
Reviewer 1 Report
First I would like to send congratulation to authors for the manuscript theme. It is of great importance.
The introduction is clearly and concisely written and finely leads to the subject of research problem.
The research method is good. Only, the authors could emphasize the response rate in the population of parents/children (addition to paragraph 93).
The results are clearly presented and the discussion follows the results.

Author Response
Please see the attachment
Dear #1 Reviewer
Thank you for your time and comments.
We attach a file with the response to your comments, as well as the new manuscript that you can find below the responses.
We remain at your disposal for any clarification in this regard.

Reviewer 2 Report
The manuscript is well written.
The keywords should be different from the words in the title. I would replace (if necessary) "comorbidity; depression; aggression" with other relevant words. This optimizes the search for the publication via search engines.
Introduction
The synthesis of the literature is good. The introduction follows a logical line starting from what is known, to what is unknown up to the hypothesis to be tested.
The gap in the literature to be filled is clearly described.
Materials and Methods
The methodology is clearly explained. The procedures are correct and the measurement tools adequate for the purpose. The statistics are appropriate.
Results
The results are clear and correctly described.
Discussion
The discussions clearly explain what was found. The take-home message is clear. The limitations of the study and future implications are clearly expressed.
Bibliographic references are adequate.
Author Response
Please see the attachment
Dear #2 Reviewer
Thank you for your time and comments.
We attach a file with the response to your comments, as well as the new manuscript that you can find below the responses.
We remain at your disposal for any clarification in this regard.

Reviewer 3 Report
The authors construct and test three structural equation models with longitudinal, self-report data on depression and aggression, gathered from children and adolescents on three separate, approximately one year time intervals (N = 106). The investigated models differed in terms of hypothesized episodic or concurrent comorbidity between depression and aggression and whether comorbidity may be a narrow band or broadband phenomenon.
The formulation of the models and the longitudinal nature of the study are laudable; the implementation of the research is, in my estimation, deficient. There is no evidence that the measuring instruments actually measure the key model constructs (depression, aggression) among participants in Spain; the study involved as single method (self-report); the fit indices have no stated justification and even if appropriate, provide only marginal support for two of the hypothesized models.
Areas of Concern:
- No psychometric data presented on the reliability and validity of the depression and aggression measures; of concern because the instruments were developed in the United States and require cross-cultural reliability and validity corroboration.
- Even if the measuring instruments have adequate psychometric characteristics, there is still the potential for method error in the use of self-report data alone.
- The aggression measure was apparently confined to two subtypes only; physical and verbal, whereas aggression appears to be a more multifaceted construct (e.g., relational; passive versus active forms of aggression).
- No consideration of possible gender differences in the conflation of depression and aggression; an important notion in relation to known disparities between boys and girls in type and/or severity of aggression and depression.
- No apparent recognition of the controversial or ethical issues involved in the selection of fit indices (see Stone, 2021. The ethical use of fit indices in structural equation modeling, Frontiers in Psychology, 12, 1-4).
- Low sample size of 106 (the standard being 200 or above)
- No rationale or justification for fit indices employed:
Consider that fit indices should be selected and justified in relation to (a) the type of estimation procedure, (b) sample size, and (c), whether there will be more than one SEM model for comparison, and (d), whether the SEM involves confirmatory factor analysis.
Specific questions for authors. Why was CFI selected over the Tucker Lewis Index (TLI)? Why was the “P of close fit (PCLOSE)” not used as a test of RMSEA? Why are GFI and AGFI generally not recommended fit indices? What does a Hoelter’s Critical N of 251 mean when the N of the study was 106?
I presume that the “Comparative Adjustment Index,” p. 3, line 130, is the “Comparative Fit Index.”
- Of the reported fit indices, chi square for all three models indicates inadequate model fit, the RMSEA is at best marginal, and the significant GFI and the AGFI are almost universally considered unacceptable indices (see Sharma, S. 2005. A simulation study to investigate the use of cutoff values for assessing model fit in covariance structure models. Journal of Business Research, 58, 935-943).
Author Response
Please see the attachment
Dear #3 Reviewer
Thank you for your time and comments.
We attach a file with the response to your comments, as well as the new manuscript that you can find below the responses.
We remain at your disposal for any clarification in this regard.

Round 2
Reviewer 3 Report
The alpha coefficients reported for the measurements are adequate; however, no data were presented in relation to the validity of these instruments.
Author Response
Dear reviewer.
I am pleased to send you in the attached file, the changes made following your suggestions.
We remain at your disposal for any clarification in this regard.
Kinds regards.
